# Deep learning via LSTM models for COVID-19 infection forecasting in India

**Rohitash Chandra**[1]*, **Ayush Jain**[2], **Divyanshu Singh Chauhan**[3]

**1** Transitional Artificial Intelligence Research Group, School of Mathematics and Statistics, University of New South Wales, Sydney, Australia, **2** Department of Electronics and Electrical Engineering, Indian Institute of Technology Guwahati, Assam, India, **3** Department of Mechanical Engineering, Indian Institute of Technology Guwahati, Assam, India

☯ These authors contributed equally to this work.
* rohitash.chandra@unsw.edu.au

**Data Availability Statement:** Data and open source code in Python is available for further analysis: https://github.com/sydney-machine-learning/LSTM-COVID-19-India.

**Funding:** The author(s) received no specific funding for this work.

## Abstract

The COVID-19 pandemic continues to have major impact to health and medical infrastructure, economy, and agriculture. Prominent computational and mathematical models have been unreliable due to the complexity of the spread of infections. Moreover, lack of data collection and reporting makes modelling attempts difficult and unreliable. Hence, we need to re-look at the situation with reliable data sources and innovative forecasting models. Deep learning models such as recurrent neural networks are well suited for modelling spatiotemporal sequences. In this paper, we apply recurrent neural networks such as long short term memory (LSTM), bidirectional LSTM, and encoder-decoder LSTM models for multi-step (short-term) COVID-19 infection forecasting. We select Indian states with COVID-19 hotpots and capture the first (2020) and second (2021) wave of infections and provide two months ahead forecast. Our model predicts that the likelihood of another wave of infections in October and November 2021 is low; however, the authorities need to be vigilant given emerging variants of the virus. The accuracy of the predictions motivate the application of the method in other countries and regions. Nevertheless, the challenges in modelling remain due to the reliability of data and difficulties in capturing factors such as population density, logistics, and social aspects such as culture and lifestyle.

## 1 Introduction

The *coronavirus disease 2019* (COVID-19) is an infectious disease caused by *severe acute respiratory syndrome coronavirus 2* (SARS-CoV-2) [1–3] which became a global pandemic [4]. COVID-19 was first identified in December 2019 in Wuhan, Hubei, China with the first confirmed (index) case was traced back to 17th November 2019 [5]. The COVID-19 pandemic forced many countries to close their borders and enforced a partial or full lockdown which had a devastating impact on the world economy [6–8]. Other major impact has been on agriculture [9, 10] which is a major source of income for the population in rural areas, especially in the developing world. The sudden lockdown in some countries created a number of problems, especially for low income communities [11, 12], and migrant workers [13]. Currently (8th

**Competing interests:** The authors have declared that no competing interests exist.

October, 2021) [14], more than 238 million cases have been reported across the world which resulted in more than 4.8 million deaths, and about 215 million have recovered [15, 16]. A significant portion of the recovered suffer from "long-covid" [17, 18], a term which refer to prolonged health problems which can last for months to entire lifetime. In terms of vaccinations (8th October, 2021) [14], 46.3% of the world population has received at least one dose and 6.4 billion doses have been administered globally. Furthermore, about 2.4% of population in low-income countries have received at least one dose.

The case of India has been unique when it comes to management of COVID-19 pandemic [13]. The first COVID-19 case in India was reported on 30 January 2020. India had two major waves of infections, where the first wave was from April to October 2020 and the second wave was from February to June 2021 [19] India currently (8th October, 2021) [14, 15] has 33,935,309 confirmed cases with 450,408 deaths which makes the largest in Asia, the second highest in the world after the United States. The fatality rate of COVID-19 in India is among the lowest in the world as it ranks 124th with 322 deaths in a million people. In comparison, United States has 2,197 and the United Kingdom has 2,013 deaths in a million people. The second wave (2021) had a devastating effect on the Indian health system and during the time around its peak in daily cases, India had highest daily infection in the world [14, 15]. On the bright side, India also has one of the fastest recovery rates in the world with 236,610 active cases, and ranks 9th in the world although 2nd in total cases.

In terms of COVID-19 forecasting, prominent computational and statistical models have been unreliable due to the complexity of the spread of infections [20–22]. Given lack of data, it is challenging to develop a model while taking into consideration population density, effect of lock-downs, effect of viral mutations and variants such as the delta variant [23], logistics and travel, and qualitative social aspects such culture and lifestyle [24]. However, culture and lifestyle are examples of variable of interest that cannot be measured quantitatively. Due to qualitative nature of certain variables of interest such as lifestyle, and lack of data collection, modelling attempts have been mostly unreliable [25]. We need to re-look at the situation with latest data sources and most comprehensive forecasting models [26–28]. Moreover, a number of other limitations exists, such as noisy or unreliable data of active cases [29], changing mortality rate given different variants, and asymptotic carriers [30, 31]. There have been reports that the models lack a number of limitations and failed in several situations [25]. Despite these challenges, it has been shown that country based mitigation factors in terms of lockdown level and monitoring has a major impact on the rate of infection [32]. We note that limited work has been done using deep learning-based forecasting models for COVID-19 in India, although the country has one of world's largest population with highly populated and highly dense cities. There is a need to evaluate latest deep learning models for forecasting COVID-19 in India, that takes into account both the first (2020) and the second wave (2021) of infections.

Deep learning models such as recurrent neural networks (RNNs) are well suited for modelling spatiotemporal sequences [33–37] and modeling dynamical systems, when compared to simple neural networks [38–40]. The limitation in training RNNs for long-term dependencies in sequences where data sequences span hundreds or thousands of time-steps [41, 42] have been addressed by *long short-term memory* networks (LSTMs) [35]. LSTMs have been used for COVID-19 forecasting in China [43] with good performance results when compared to epidemic models. LSTMs have also been used for COVID-19 forecasting in Canada [44]. Other deep learning models such as convolutional neural networks (CNNs) have recently shown promising performance for time series forecasting [45, 46]. Hence, they would also be suited for capturing spatiotemporal relationship of COVID-19 transmission with neighbouring states in India.

In this paper, we employ LSTM models in order to forecast the spread of COVID-infections among selected states in India. We select Indian states with COVID-19 hotpots and capture the first and second wave of infections in order to later provide two months ahead forecast. We first employ univariate and multivariate time series forecasting approaches and compare their performance for short-term (4 days ahead) forecasting. We also present visualisation and analysis of the COVID-19 infections and provide open source software framework that can provide robust predictions as more data gets available. The software framework can also be applied to different countries and regions.

The rest of the paper is organised as follows. Section 2 presents a background and literature review of related work. Section 3 presents the proposed methodology with data analysis and Section 4 presents experiments and results. Section 5 provides a discussion with discussion of future work and Section 6 concludes the paper.

## 2 Related work

The pandemic has greatly affected and transformed work environment and lifestyle. COVID-19 lockdowns and restrictions of movement has given rise to e-learning [47–49] and telemedicine [50], and created opportunities in applications for geographical information systems [51]. The lockdown showed positive impact on the environment [52, 53], especially for highly populated and industrial nations with high air pollution rate [54]. Zambrano-Monserrate et. al highlighted the positive indirect effects revolve around the reduction air pollutants in China, France, Germany, Spain, and Italy [52]. However, the way medical pollutants and domestic waste were discarded during lockdowns has been an issue [52]. COVID-19 lockdowns and infection management raised concerns about prejudices against minorities and people of colour in developed countries such as the United States [55]. Furthermore, there has been a significant impact on mental health across the globe [56, 57].

It has been shown that in some countries, comprehensive identification and isolation policies have effectively suppressed the spread of COVID-19. Huang et. al [58] presented an evaluation of identification and isolation policies that effectively suppressed the spread of COVID-19 which further contributed to reduce casualties during the phase of a dramatic increase in diagnosed cases in Wuhan, China. The authors recommended that governments should swiftly execute the forceful public health interventions in the initial stage of the pandemic. However, such policies have not been that effective for other countries, such as the first wave of infections and associated lockdowns in India [26].

### 2.1 Modelling and forecasting COVID-19

A number of machine learning and statistical models have been used for modelling and forecasting COVID-19 in different parts of the world. Saba and Elsheikh presented simple autoregressive neural networks for forecasting the prevalence of COVID-19 outbreak in Egypt which showed relatively good performance when compared to officially reported cases [22]. Yousaf et. al used *auto-regressive integrated moving average* (ARIMA) model for forecasting COVID-19 for Pakistan [21]. The model predicted that the number of confirmed cases would increase by factor of 2.7 giving 95% prediction interval by the end of May 2020, to 5681—33079 cases. However, Pakistan reported around 70,000 cases [14] end of May 2020, and hence the model was poor in prediction. Velásquez and Lara used Gaussian process regression model for forecasting COVID-19 infection in the United States [20]. The authors show that COVID-19 would peak in United States around July 14th 2020, with about 132,074 deaths and 1,157,796 infected individuals at the peak stage. However, the actual cases by July 14th reached more

than 3.5 million with more than 139 thousand deaths [14, 15] which shows that the model was close in forecasting deaths but forecast of total cases was poor.

Chimmula and Zhand used LSTM neural networks for time series forecasting of COVID-19 transmission in Canada [44]. The authors predicted the possible ending point of the outbreak around June 2020 and compared transmission rate of Canada with Italy and the United States. Canada reached the daily new cases peak by 2nd May 2020 [14, 15], and since then, new cases has been drastically reducing. Therefore we can say that the approach by the authors was somewhat close in reporting the peak for COVID-19 in Canada. Chakraborty and Ghosh [27] used hybrid ARIMA and wavelet-based forecasting model for short-term (ten days ahead) forecasts of daily confirmed cases for Canada, France, India, South Korea, and the United Kingdom. The authors also applied an optimal regression tree algorithm to find essential causal variables that significantly affect the case fatality rates for different countries. Maleki et. al [28] used autoregressive time series models based on mixtures of normal distribution from confirmed and recovered COVID-19 cases worldwide.

Ren et al. [24] analysed spatiotemporal variations of the epidemics before utilizing the ecological niche models with nine socioeconomic variables for identifying the potential risk zones for megacities such as Beijing, Guangzhou and Shenzhen. The results demonstrated that the method was capable of being employed as an early forecasting tool for identifying the potential COVID-19 infection risk zones. Alzahrani et al. [59] used autoregressive and ARIMA models for COVID-19 in Saudi Arabia with data till 20th April 2020 and predicted 7668 daily new cases by 21st May 2020 given stringent precautionary control measures were not implemented. However, Saudi Arabia on 21st May 2020 reported 2532 actual cases [14, 15]; hence, the model has shown poor performance. Singh et al. [60] presented a hybrid of discrete wavelet decomposition and ARIMA models in application to one month forecast the casualties cases of COVID-19 in most affected countries back then which included France, Italy, Spain, United Kingdom and and United Sates. The study found that the hybrid model was better than stand-alone models. Dasilva et al. [26] employed machine learning methods such as Bayesian regression neural network, cubist regression, k-nearest neighbors, quantile random forest, and support vector regression with pre-processing based on variational mode decomposition for forecasting one, three, and six-days-ahead the cumulative COVID-19 cases in five Brazilian and American states up to April 28th, 2020. Yang et al. [43] presented an epidemiological model that incorporated the domestic migration data and the most recent COVID-19 epidemiological data to predict the epidemic progression. The model predicted peak by late February, showing gradual decline by end of April 2020. This was one of the few attempts in prediction of COVID-19 infection trend in China [14, 15]; however, the actual peak was observed in early February 2020 and the spread of infections ended by the middle of March 2020.

Next, we review key studies of COVID-19 forecasting with deep learning models in India. Anand et al. [61] focused on forecasting of COVID-19 cases in India using RNNs such as LSTM and gated-recurrent units (GRU) with the dataset from 30th January 2020 to 21st July 2020. Bhimala et al. [62] incorporated the weather conditions of different states to make improve forecasting of the COVID-19 cases in different states of India. The authors made assumption that different humidity levels in different states will lead to varying transmission of infection within the population. They demonstrated that LSTM model performed better in the medium and long range forecasting scale when integrated with the weather data. Shetty [63] presented real-time forecasting using a simple neural network for the COVID-19 cases in the state of Karnataka in India where parameter selection for the model was based on cuckoo search algorithm. The study reported that the mean-absolute percentage error (MAPE) was reduced from 20.73% to 7.03% and the proposed model was further tested on the Hungary

COVID-19 dataset and reported promising results. Tomar and Gupta [64] developed LSTM model for 30-day ahead prediction of COVID-19 positive cases in India where they also studied the effect of preventive measures on the spread of COVID-19. They showed that with preventive measures and lower transmission rate, the spread can be reduced significantly. Gupta et al. [65] forecasted COVID-19 cases of India using support vector machines, prophet, and linear regression models. Similarly, Bodapati et al. [66] forecasted the COVID-19 daily cases, deaths caused and recovered cases with the help of LSTM networks for the whole world. Chaurasia and Pal [67] used several forecasting models such as simple average, single exponential smoothing, Holt winter method, and ARIMA models for COVID-19 pandemic.

A number of machine learning methods used in conjunction with deep learning models for COVID-19 forecasting for the rest of the world. Battineni et al. [68] forecasted COVID-19 cases using a machine learning method known as prophet logistic growth model which estimated that by late September 2020, the outbreak can reach 7.56, 4.65, 3.01 and 1.22 million cases in the United States, Brazil, India and Russia, respectively. Nadler et al. [69] used a model embedded in a Bayesian framework coupled with a LSTM network to forecast cases of COVID-19 in developed and developing countries. Istaiteh et al. [70] compared the performance of ARIMA, LSTM, multilayer perceptron and convolutional neural network (CNN) models for prediction of COVID-19 cases all over the world. They reported that deep learning models outperformed ARIMA model, and furthermore CNN outperformed LSTM networks and multi-layer perceptron. Pinter et al. [71] used hybrid machine learning methods consisting of adaptive network-based fuzzy inference systems (ANFIS) and mutlilayer perceptron (simple neural network) for COVID-19 infections and mortality rate in Hungary.

## 3 Methodology: Forecasting COVID-19 novel infections with deep learning models

We need to reconstruct the original time series into a state-space vector in order to train deep learning models for multi-step-ahead prediction. Taken's theorem expresses that the reconstruction can reproduce important features of the original time series [72]. Hence, an embedded phase space $Y(t) = [(x(t), x(t - T), . . ., x(t - (D - 1)T)]$ can be generated given an observed time series $x(t)$; where $T$ is the time delay, $D$ is the embedding dimension (window size) $t = 0$, 1, 2, . . ., $N - D - 1$, and $N$ is the length of the original time series. Appropriate values for $D$ and $T$ need to selected to efficiently apply Taken's theorem for reconstruction [73]. Taken's proved that if the original attractor is of dimension $d$, then $D = 2d + 1$ would be sufficient [72].

### 3.1 LSTM network models

Recurrent neural networks (RNNs) have been prominent for modelling temporal sequences. RNNs feature a context layer to act as memory in order to project information from current state into future states, and eventually the output layer. Although number of different RNN architectures exist, the Elman RNN [33, 74] is one of the earliest which has been prominent for modelling temporal sequences and dynamical systems [39, 75, 76].

Training RNNs in the early days has been a challenging task. Backpropagation-through-time (BPTT) which is an extension of the backpropagation algorithm has been prominent in training RNNs [34]. BPTT features gradient-descent where the error is backpropagated for a deeper network architecture that features states defined by time. The RNN unfolded in time is similar to a multilayer perceptron that features multiple hidden layers. A major limitation of BPTT for simple RNNs has been the problem of learning long-term dependencies given vanishing and exploding gradients [41]. The LSTM network addressed this limitation with better capabilities in remembering the long-term dependencies using memory cells in the hidden

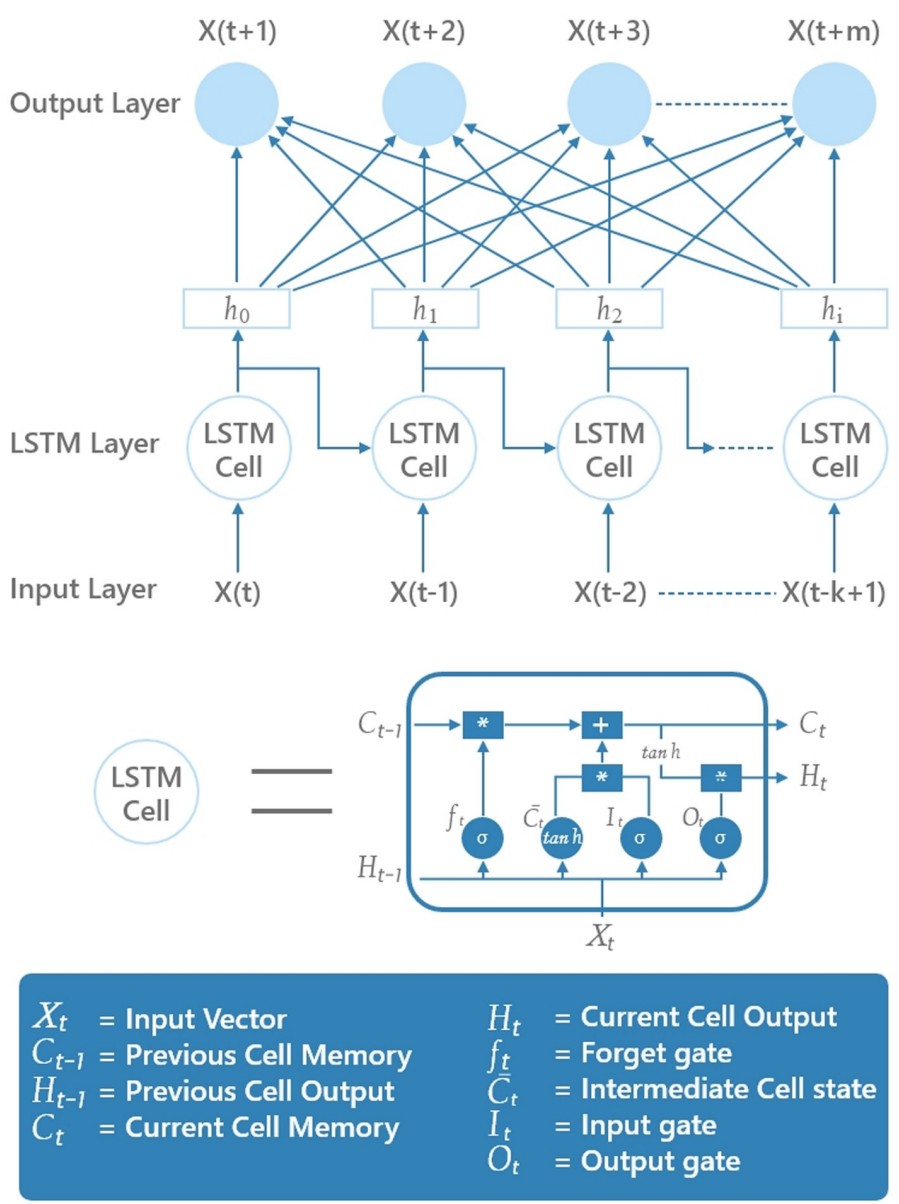

**Fig 1. LSTM memory cell in the LSTM recurrent neural network.**

layer [35]. The memory cells help in remembering the long-term dependencies in data as shown in Fig 1.

The LSTM network model calculates a hidden state output $h_t$ by

$$
\begin{aligned}
i_t &= \sigma(x_t U^i + h_{t-1} W^i) \\
f_t &= \sigma(x_t U^f + h_{t-1} W^f) \\
o_t &= \sigma(x_t U^o + h_{t-1} W^o) \\
\tilde{C}_t &= \tanh(x_t U^c + h_{t-1} W^c) \\
C_t &= \sigma(f_t * C_{t-1} + i_t * \tilde{C}_t) \\
h_t &= \tanh(C_t) * o_t
\end{aligned}
\tag{1}
$$

where, $i_t$, $f_t$ and $o_t$ refer to the input, forget and output gates, at time $t$, respectively. $c$ refers to the memory cell. $x_t$ and $h_t$ refer to the number of input features and number of hidden units, respectively. $W$ and $U$ are the weight matrices adjusted during learning along with $b$, which is the bias. Note that all the gates have the same dimensions $d_h$, which is given by the size of hidden state. $\tilde{C}_t$ is the intermediate cell state, and $C_t$ is the current cell memory. The initial values at $t = 0$ are given by $C_0 = 0$ and $h_0 = 0$. Note that we denote ($^*$) as element-wise multiplication.

### 3.2 Bi-directional LSTM networks

Conventional RNN and LSTM networks only make use of previous context state for determining future states. Bidirectional RNNs (BD-RNNs) [77] on the other hand, process information in both directions with two separate hidden layers which are then propagated forward to the same output layer. Hence, two independent RNNs are placed together to allow both backward and forward information about the sequence at every time step. The forward hidden sequence $h_f$, the backward hidden sequence $h_b$, and the output sequence $y$ are computed by iterating the backward layer from $t = T$ to $t = 1$, and the forward layer from $t = 1$ to $t = T$.

Bi-directional LSTM networks (BD-LSTM) [78] can access longer-range context or state in both directions similar to BD-RNNs. BD-LSTM networks were originally proposed for word-embedding in natural language processing [78] tasks and have been used in several real-world sequence processing problems such as phoneme classification [78], continuous speech recognition [79], and speech synthesis [80].

BD-LSTM networks intake inputs in two ways; one from past to future, and another from future to past by running information backwards so that state information from the future is preserved. Given two hidden states combined in any point in time, the network can preserve information from both past and future as shown in Fig 2.

### 3.3 Encoder-decoder LSTM networks

The encoder-decoder LSTM network (ED-LSTM) [81] was introduced as a sequence to sequence model for mapping a fixed-length input to a fixed-length output. The length of the input and output may differ which makes them applicable in automatic language translation tasks (English to French for example) which can be extended to multi-step series prediction where both the input and outputs are of variable lengths. A latent vector representation is used to handle variable-length input and outputs by first encoding the input sequences, one at a time and then decoding it. We consider the input sequence $(x_1, \ldots, x_n)$ with corresponding output sequence $(y_1, \ldots, y_m)$, and estimate the conditional probability of the output sequence given an input sequence, i.e. $p(y_1, \ldots, y_m | x_1, \ldots, x_n)$. In the encoding phase, given an input sequence, the ED-LSTM network computes a sequence of hidden states. In the decoding phase, it defines a distribution over the output sequence given the input sequence as shown in Fig 3.

### 3.4 India: Situation report: 8th October, 2021

We provide a visual representation of the total number of COVID-19 infections for different states and union territories in India based on data till 8th October, 2021.

Tables 1–4 provide a rank of top ten Indian states with total cases 1st of every month. We see that largely populated states such as Maharashtra (population estimate of 123 million [82]) has been leading India in number of total cases through-out 2020. We note that state of Uttar Pradesh has estimate population of 238 million has managed better. Delhi has a relatively smaller population (estimated 19 million [82]), but high population density and hence been

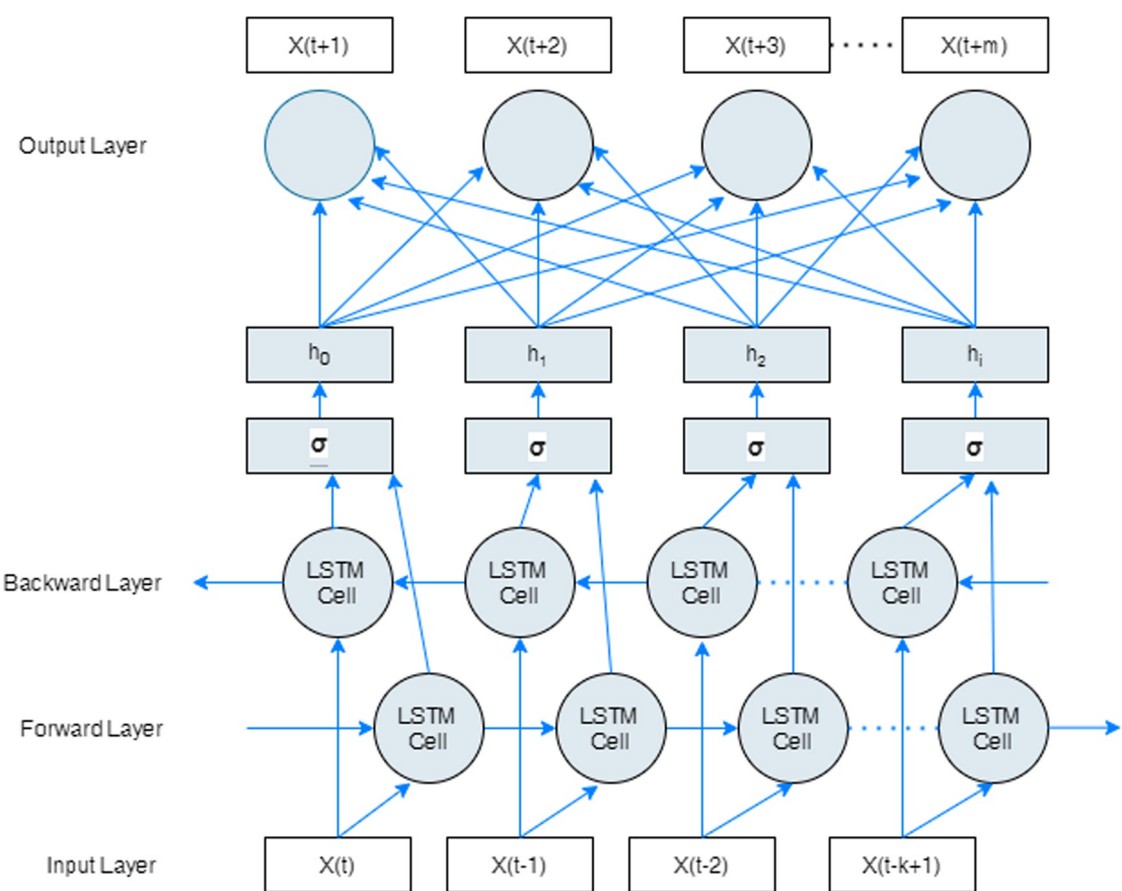

**Fig 2. Bi-directional LSTM recurrent neural network.**

one of the leading states with COVID-19 infections (in top 6 throughout 2020). Tables 3 and 4 show that in 2021, Maharashtra continued leading; however, the second position was overtaken by Kerala from February which maintained the position since then. We note that from February to June 2021, India experienced the second-wave of infections from the delta-variant of the virus, with Maharastra and Kerala leading most of the time in terms of monthly infections. The first peak for novel cases in India was reached on September 16th 2020 with 97,894 daily and 93,199 weekly average novel cases [15]. The daily novel cases were steady for several months and then raised again from February 2021 for the second wave of infections. The peak of the second wave was reached around 7th May 2021, with 401,078 daily and weekly average of 389,672 novel infections [15]. The peak of deaths was reached around 21st May 2021 with 4194 deaths and 4188 weekly average.

Figs 4 and 5 present the total number of novel weekly cases for different groups of Indian states and union territories, which covers both the first and second wave of infections. We notice that the number of cases significantly increased after May 2020 which marks the first wave and then declined. Fig 4 (Panel a) focusing on major affected states show that Maharastra led the first and second wave of infections followed by Karnataka. In Fig 4 Panel (b), considering the Eastern states, we find that new cases in West Bengal drastically increased for the first wave of infections and it took Bihar longer to reach the peak when compared to Odisha and the others. In the second wave, West Bengal led the other states by a large margin. In the

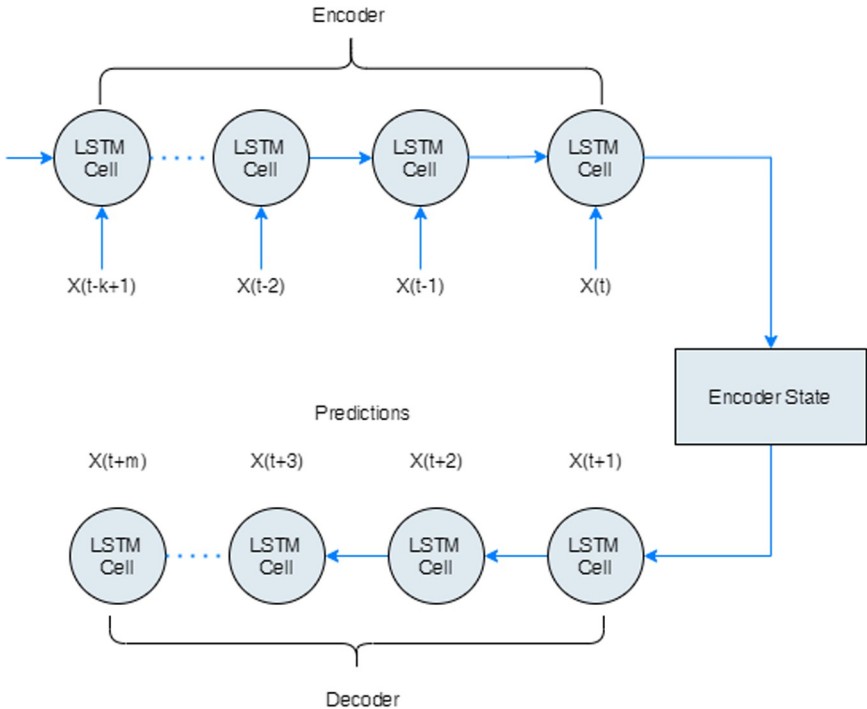

**Fig 3. Encoder-decoder LSTM recurrent neural network.** Note that the outputs $(y_1, \ldots, y_m)$ with corresponding input $(x_1, \ldots, x_n)$ are not explicitly shown.

**Table 1. Rank of states by number of novel total cases taken at the 1st of every month for April to August 2020 [15].** The number of novel cases are shown in brackets.

| Rank | April | May | June | July | August |
|---|---|---|---|---|---|
| 1 | Maharashtra | Maharashtra | Maharashtra | Maharashtra | Maharashtra |
| | (302) | (10498) | (67655) | (174761) | (422118) |
| 2 | Kerala | Gujarat | Tamil Nadu | Tamil Nadu | Tamil Nadu |
| | (241) | (4395) | (22333) | (90167) | (245859) |
| 3 | Tamil Nadu | Delhi | Delhi | Delhi | Andhra Pradesh |
| | (234) | (3515) | (19844) | (87360) | (140933) |
| 4 | Delhi | Madhya Pradesh | Gujarat | Gujarat | Delhi |
| | (152) | (2719) | (16779) | (32557) | (135598) |
| 5 | Uttar Pradesh | Rajasthan | Rajasthan | Uttar Pradesh | Karnataka |
| | (103) | (2584) | (8831) | (23492) | (124115) |
| 6 | Karnataka | Tamil Nadu | Madhya Pradesh | West Bengal | Uttar Pradesh |
| | (101) | (2323) | (8089) | (18559) | (85461) |
| 7 | Telengana | Uttar Pradesh | Uttar Pradesh | Rajasthan | West Bengal |
| | (96) | (2281) | (7823) | (18014) | (70188) |
| 8 | Rajasthan | Andhra Pradesh | West Bengal | Telengana | Telengana |
| | (93) | (1463) | (5501) | (16339) | (62703) |
| 9 | Andhra Pradesh | Telengana | Bihar | Karnataka | Gujarat |
| | (83) | (1039) | (3815) | (15242) | (61438) |
| 10 | Gujarat | West Bengal | Andhra Pradesh | Andhra Pradesh | Bihar |
| | (82) | (795) | (3679) | (14595) | (51233) |

**Table 2. Rank of states by number of novel total cases taken at the 1st of every month for September to December 2020 [15].** The number of novel cases are shown in brackets.

| Rank | September | October | November | December |
|---|---|---|---|---|
| 1 | Maharashtra | Maharashtra | Maharashtra | Maharashtra |
| | (792541) | (1384446) | (1678406) | (1823896) |
| 2 | Andhra Pradesh | Andhra Pradesh | Karnataka | Karnataka |
| | (434771) | (693484) | (823412) | (884897) |
| 3 | Tamil Nadu | Karnataka | Andhra Pradesh | Andhra Pradesh |
| | (428041) | (601767) | (823348) | (868064) |
| 4 | Karnataka | Tamil Nadu | Tamil Nadu | Tamil Nadu |
| | (342423) | (597602) | (724522) | (781915) |
| 5 | Uttar Pradesh | Uttar Pradesh | Uttar Pradesh | Kerala |
| | (230414) | (399082) | (481863) | (602982) |
| 6 | Delhi | Delhi | Kerala | Delhi |
| | (174748) | (279715) | (43310) | (570374) |
| 7 | West Bengal | West Bengal | Delhi | Uttar Pradesh |
| | (162778) | (257049) | (386706) | (543888) |
| 8 | Bihar | Odisha | West Bengal | West Bengal |
| | (136457) | (219119) | (373664) | (483484) |
| 9 | Telengana | Kerala | Odisha | Odisha |
| | (127697) | (196106) | (290116) | (318725) |
| 10 | Assam | Telengana | Telengana | Telengana |
| | (109040) | (193600) | (240048) | (270318) |

**Table 3. Rank of states by number of novel total cases taken at the 1st of every month for January to May 2021 [15].** The number of novel cases are shown in brackets.

| Rank | January | February | March | April | May |
|---|---|---|---|---|---|
| 1 | Maharashtra | Maharashtra | Maharashtra | Maharashtra | Maharashtra |
| | (2026399) | (2155070) | (2812980) | (4602472) | (5746892) |
| 2 | Karnataka | Kerala | Kerala | Kerala | Karnataka |
| | (939387) | (1059403) | (1124584) | (1571183) | (2604431) |
| 3 | Kerala | Karnataka | Karnataka | Karnataka | Kerala |
| | (929178) | (951251) | (997004) | (1523142) | (2526579) |
| 4 | Andhra Pradesh | Andhra Pradesh | Andhra Pradesh | Uttar Pradesh | Tamil Nadu |
| | (887836) | (889916) | (901989) | (1252324) | (2096516) |
| 5 | Tamil Nadu | Tamil Nadu | Tamil Nadu | Tamil Nadu | Andhra Pradesh |
| | (838340) | (851542) | (886673) | (1166756) | (1693085) |
| 6 | Delhi | Delhi | Delhi | Delhi | Uttar Pradesh |
| | (635096) | (639289) | (662430) | (1149333) | (1691488) |
| 7 | Uttar Pradesh | Uttar Pradesh | Uttar Pradesh | Andhra Pradesh | Delhi |
| | (600299) | (603527) | (617194) | (1101690) | (1426240) |
| 8 | West Bengal | West Bengal | West Bengal | West Bengal | West Bengal |
| | (569998) | (575118) | (586915) | (828366) | (1376377) |
| 9 | Odisha | Odisha | Chhattisgarh | Chhattisgarh | Chhattisgarh |
| | (335072) | (337191) | (349187) | (728700) | (971463) |
| 10 | Rajasthan | Rajasthan | Odisha | Rajasthan | Rajasthan |
| | (317491) | (320336) | (340917) | (598001) | (939958) |

**Table 4. Rank of states by number of novel total cases taken at the 1st of every month for June to September 2021 [15].** The number of novel cases are shown in brackets.

| Rank | June | July | August | September |
|---|---|---|---|---|
| 1 | Maharashtra | Maharashtra | Maharashtra | Maharashtra |
| | (6061404) | (6303715) | (6464876) | (6541119) |
| 2 | Kerala | Kerala | Kerala | Kerala |
| | (2924165) | (3390761) | (4057233) | (4613937) |
| 3 | Karnataka | Karnataka | Karnataka | Karnataka |
| | (2843810) | (2905124) | (2949445) | (2972620) |
| 4 | Tamil Nadu | Tamil Nadu | Tamil Nadu | Tamil Nadu |
| | (2479696) | (2559597) | (2614872) | (2655572) |
| 5 | Andhra Pradesh | Andhra Pradesh | Andhra Pradesh | Andhra Pradesh |
| | (1889513) | (1966175) | (2014116) | (2045657) |
| 6 | Uttar Pradesh | Uttar Pradesh | Uttar Pradesh | Uttar Pradesh |
| | (1706107) | (1708441) | (1709335) | (1709761) |
| 7 | West Bengal | West Bengal | West Bengal | West Bengal |
| | (1499783) | (1528019) | (1548604) | (1565645) |
| 8 | Delhi | Delhi | Delhi | Delhi |
| | (1434188) | (1436265) | (1437764) | (1438685) |
| 9 | Chhattisgarh | Chhattisgarh | Odisha | Odisha |
| | (994480) | (1002008) | (1007750) | (1023735) |
| 10 | Rajasthan | Odisha | Chhattisgarh | Chhattisgarh |
| | (952422) | (977268) | (1004451) | (1005229) |

Northern states, shown in Fig 5 (Panel a), we find that Uttar Pradesh leading the first and second wave of infections which is not surprising since it is the most populous state of India. In the case of the relatively lowly populated states (small states) shown in Fig 5 (Panel b), we find that Goa and Tripura lead the first wave of infections and later in the second wave, Goa overtakes the rest of the states, significantly.

Fig 6 presents daily active cases and cumulative (total) deaths for key Indian states for 2021 [15]. We notice that the different states, such as Maharastra and Tamil Nadu have few to several weeks of lag in reaching the peak of novel daily cases. In terms of deaths, we do not see a sharp increase after July 2021 in most of states. Note that we chose not to show daily deaths in the same graphs since the scales between active cases and deaths are quite different.

## 4 Results

In this section, we present results of prediction of COVID-19 daily cases in India using prominent LSTM neural network models that includes, BD-LSTM and ED-LSTM with architectural details given earlier (Section 3).

### 4.1 Experimental design

Our experiments consider the evaluation of the respective models for univariate and multivariate prediction tasks. The data has been accessed from Indian Institute of Statistical Science—Bangalore [83], which was originally sourced from Ministry of Health and Family Welfare, Government of India website [84]. The dataset is based on daily novel cases which is normalised taking the maximum number of daily cases over the entire data into account. We start our analysis from 15th April 2020 and use rolling mean of 3 days to smoothen the original data.

(a)

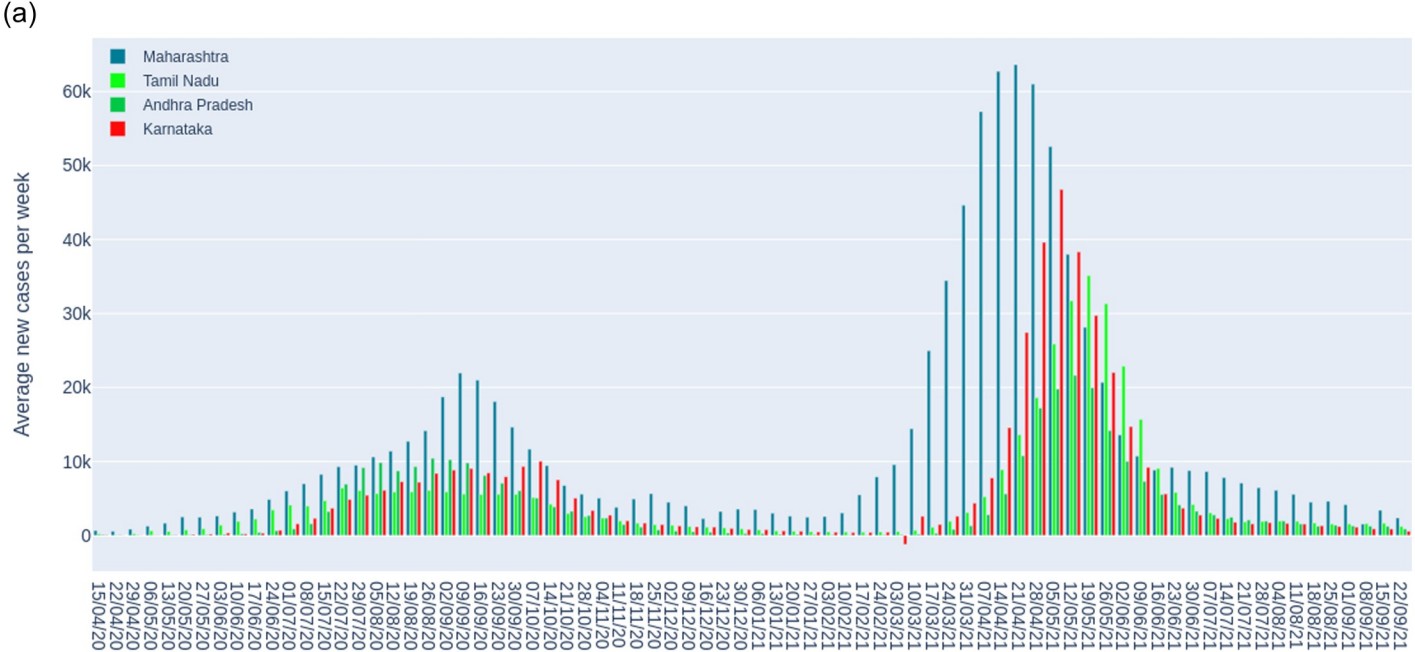

(b)

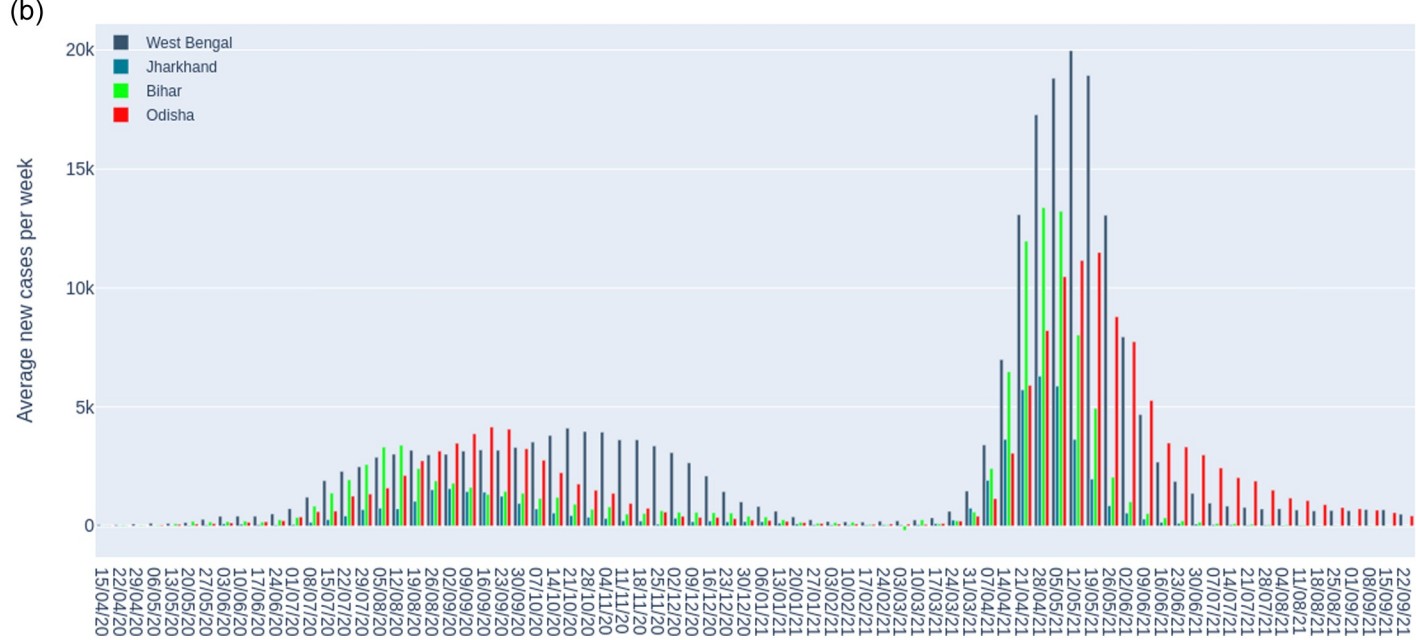

**Fig 4. Weekly average of new cases for groups of Indian states and union territories [15].**

We reconstruct the univariate and multivariate time series into a state-space vector using Taken's theorem [72] with selected values for embedding dimension window ($D = 6$) and time-lag ($T = 2$) for multi-step ahead (MSA) prediction. We consider four prediction horizons; i.e. $MSA = 4$, where each step is a prediction horizon.

The Adam optimizer is used for training the respective LSTM models. Tables 5 and 6 describe the topology for the respective LSTM models for univariate and multivariate cases,

(a)

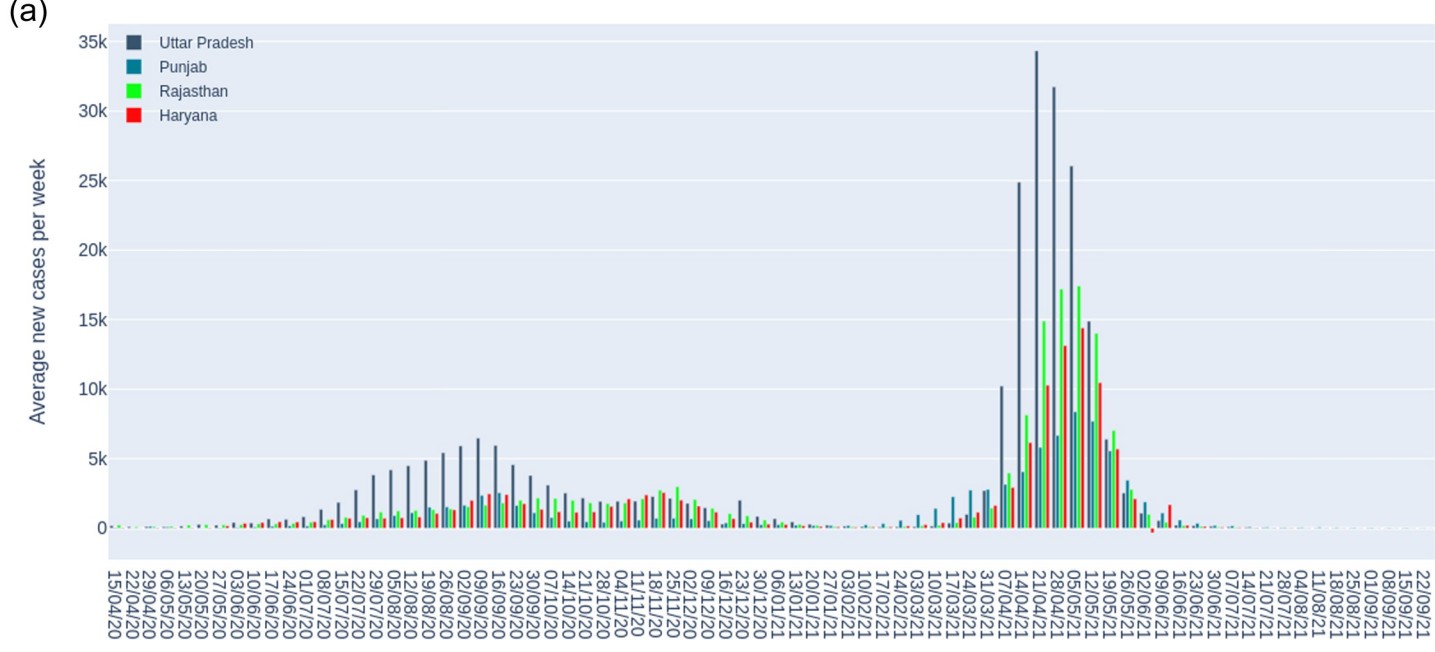

(b)

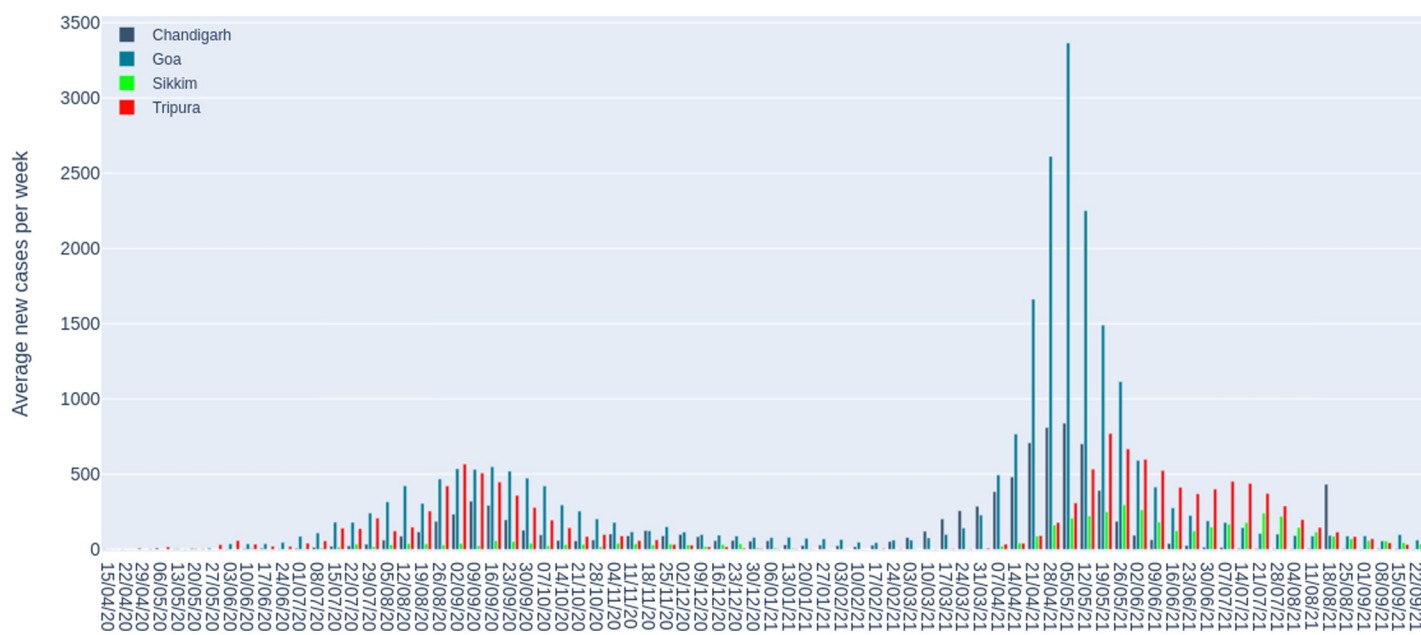

**Fig 5. Weekly average of new cases for groups of Indian states and union territories [15].**

respectively. In case of multivariate model, the input contains four features which represents the adjacent states in relation to the state taken into account; i.e. the case of Maharashtra (Maharashtra, Gujarat, Madhya Pradesh, Uttar Pradesh) and the case of Delhi (Delhi, Rajasthan, Uttar Pradesh, Haryana). In multivariate model of India, we take all the states as input features to the multivariate model. We note that similar to univariate model, the multivariate

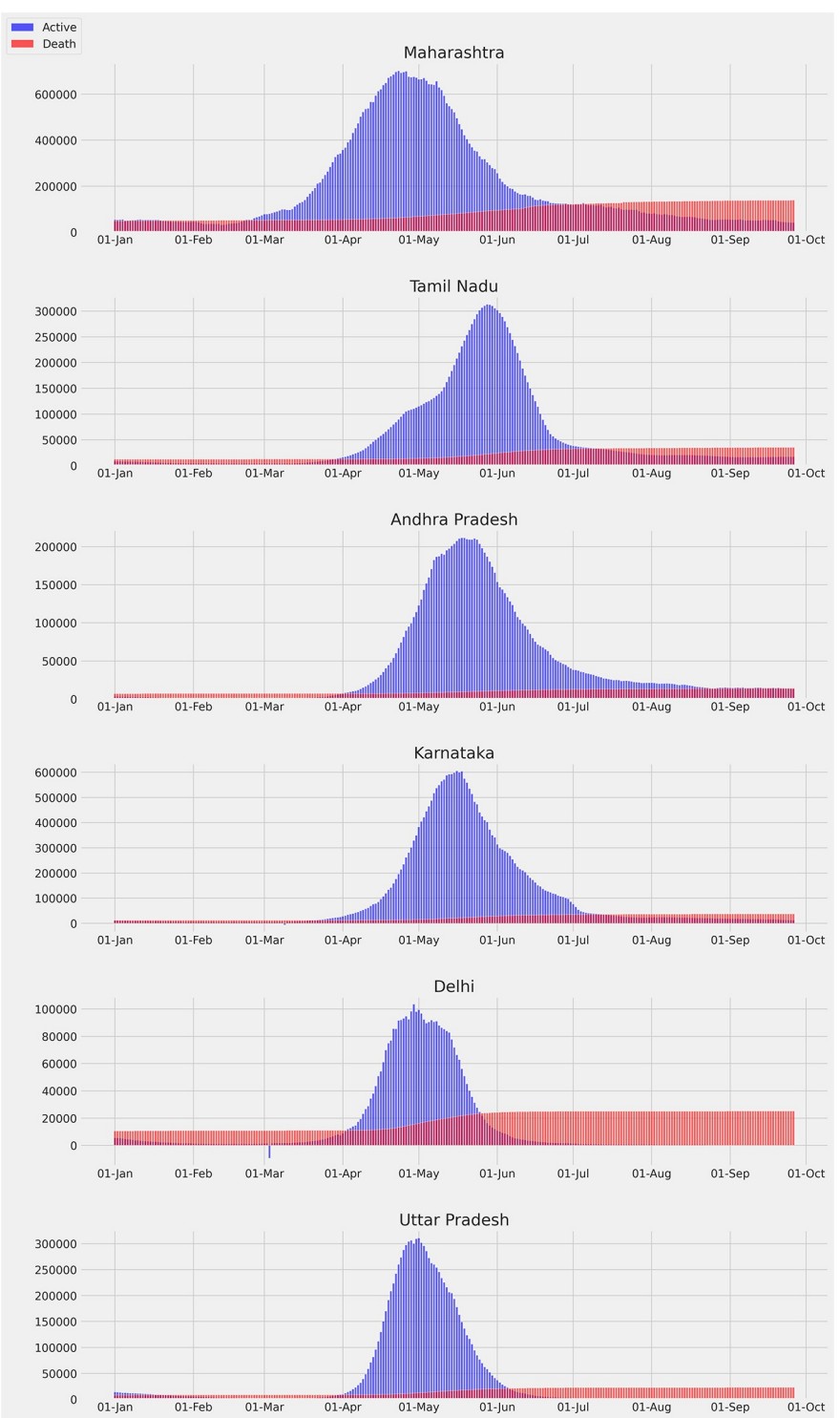

**Fig 6. Daily active cases and cumulative death in key states of India for 2021 [15].**

**Table 5. Respective LSTM model topologies for the univariate case.**

| Method | Input | Hidden layer 1 | Hidden layer 2 | Output |
|---|---|---|---|---|
| LSTM | (6,1) | 32 | 32 | (1,4) |
| BD-LSTM | (6,1) | 32 | 16 | (1,4) |
| ED-LSTM | (6,1) | 32 | - | (1,4) |

model considers a selected embedding dimension window ($D = 6$) for multi-step-ahead prediction ($MSA = 4$). Tables 5 and 6 provides the details for the respective LSTM model typologies in terms of the hidden neurons and layers.

We review the performance of the respective methods in terms of scalability and robustness which refers to the ability of maintaining consistent prediction performance as the prediction horizon increases. We use the root mean squared error (RMSE) in Eq 2 as the main performance measure for prediction accuracy

$$RMSE = \sqrt{\frac{1}{N}\sum_{i=1}^{N}(y_i - \hat{y}_i)^2} \tag{2}$$

where $y_i, \hat{y}_i$ are the observed data, predicted data, respectively. $N$ is the length of the observed data. We use RMSE for each prediction horizon and for each problem, we report the mean error for the respective prediction horizons.

We present the mean and 95% confidence interval for 30 experiment runs with different initialisation of model parameter space (weights and biases) in all the experiments. We use a dropout rate of 0.2 for the respective models in all the experiments.

## 4.2 Prediction performance

We first evaluate the optimal strategy for creating training and testing datasets. We use static-split of training samples from 15th April 2020 to 15th May 2021. Our test set features data from 16th May 2021 to 27th September 2021; hence, the training data covers half of the second wave of the cases. In random-split, we create the train and test sets by randomly shuffling the dataset with the same size of the dataset as done for the static-split. We show results for entire case of India, and two leading states of COVID-19 infections, i.e. Maharashtra and Delhi. We investigate the effect of the univariate and multivariate approaches on the three models, (LSTM, BD-LSTM, ED-LSTM). Finally, using the best model, we provide a two month outlook for novel daily cases with a recursive approach, i.e. by feeding back the predictions into the trained models.

Fig 7 shows univariate LSTM, BD-LSTM and ED-LSTM models with a static-split of train/test dataset. Fig 8 shows univariate random splitting of train/test datasets using the same models. We observe that the prediction for the India dataset has a unique trend where the model is improving with increase in the prediction horizon (steps) when compared to Maharashtra and Delhi cases (Panels d and f). The corresponding cases in random-split given in Fig 8 show a

**Table 6. Respective LSTM model topologies for the multivariate case.**

| Method | Input | Hidden-layer | Output |
|---|---|---|---|
| LSTM | (6,4) | 32 | (1,4) |
| BD-LSTM | (6,4) | 32 | (1,4) |
| ED-LSTM | (6,4) | 32 | (1,4) |

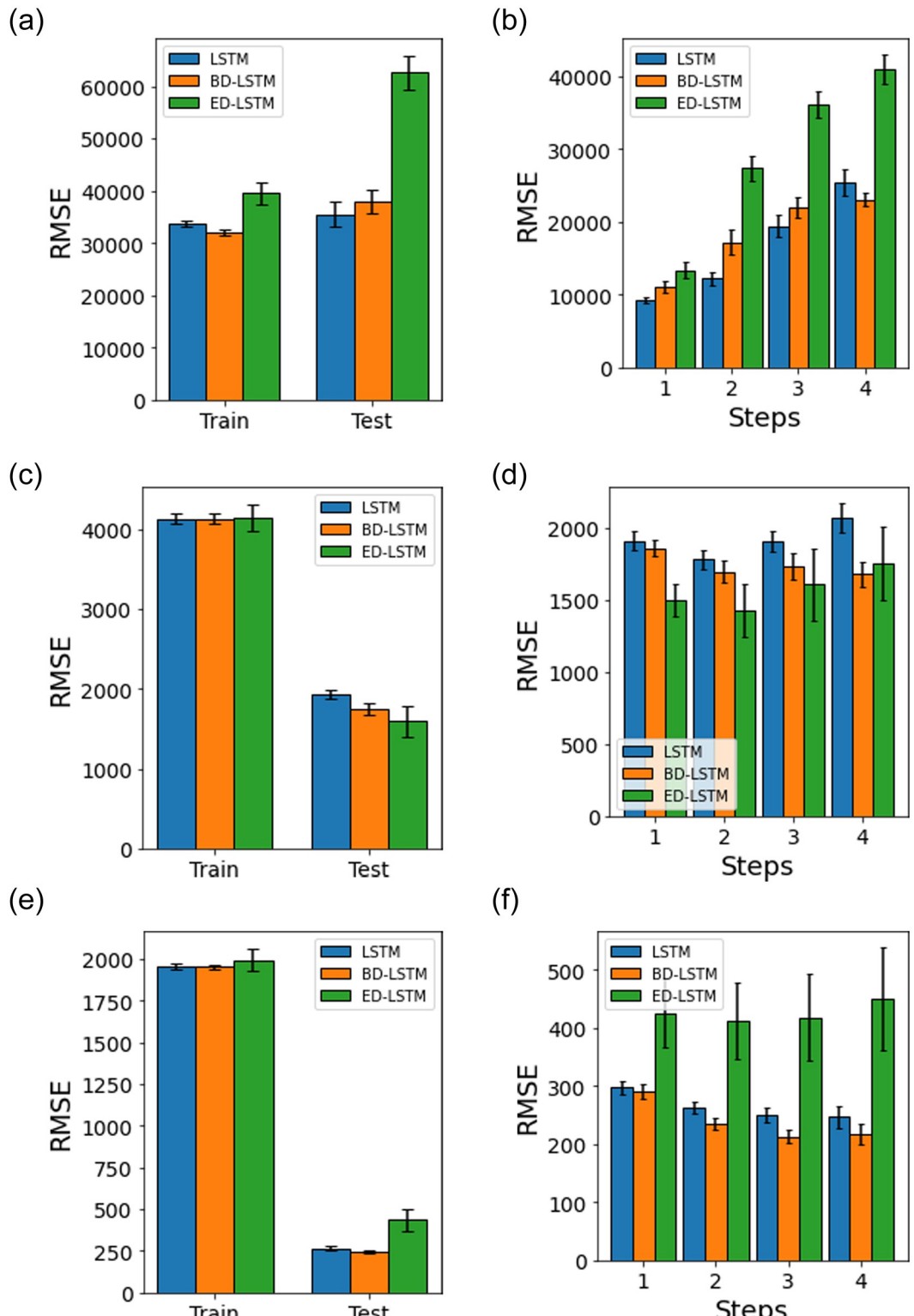

**Fig 7. Univariate LSTM, BD-LSTM and ED-LSTM model performance accuracy (RMSE) with a static-split for the train and test datasets, and test prediction horizons (steps).** The error bars represent the mean and 95% confidence interval for 30 experiment runs.

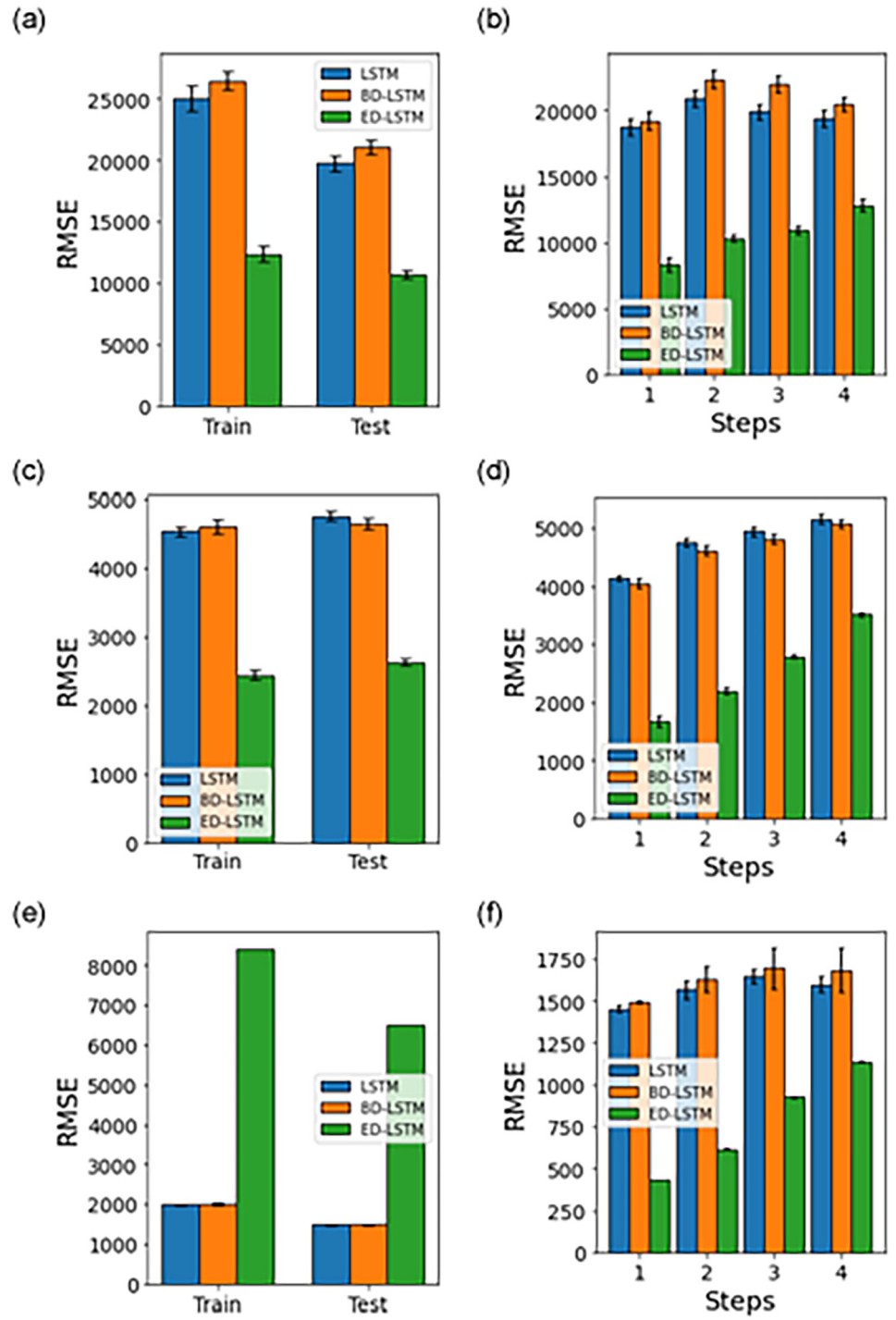

**Fig 8. Univariate LSTM, BD-LSTM and ED-LSTM model performance accuracy (RMSE) with a random-split for the train and test datasets, and the test prediction horizons (steps).** The error bars represent the mean and 95% confidence interval for 30 experiment runs.

different trend and better accuracy (with lower RMSE) where ED-LSTM provides the best test accuracy. In general, we find that random-split with ED-LSTM provides the best performance accuracy for the given univariate models.

Fig 9 shows results for the multivariate approach, where we use the same methods used in the univariate approach (LSTM, ED-LSTM, BD-LSTM). We find that ED-LSTM model gives best performance for the test datasets for all the respective datasets. Fig 10 shows results for the case of random shuffling of train/test dataset using the respective methods. We notice that ED-LSTM at times, provides slightly worse performance when compared to BD-LSTM. More-over, the performance is not as good when compared to static-split (Fig 9) and in general, we find that ED-LSTM with static-split provides the best performance accuracy.

Tables 7 and 8 provide a summary of results in terms of the test dataset performance accuracy (RMSE) by the respective models for random-split and static-split, which have been given in Figs 7–10. In the univariate models, ED-LSTM provides the best performance accuracy across most of the three different datasets while LSTM performs best only for a single case (Fig 7, Panel a). In multivariate models, BD-LSTM and ED-LSTM provide the best performance accuracy for most cases while LSTM performs best only for a single case (Fig 10, Panel a).

Next, we select two univariate recursive models using random-split for the three datasets to provide a two months outlook for COVID-19 daily infections. In this approach, we use the predictions using the test dataset and extend it further for two months (October and November 2021), recursively. Fig 11 presents results for univariate LSTM and BD-LSTM models. The uncertainty (95% confidence interval shaded in green) and mean prediction is shown in solid black line for 30 experiment runs. We notice that there is a trend of general decline in cases and we also find that the LSTM models well capture the spike and fall in cases every few days.

## 5 Discussion

The COVID-19 pandemic in India was hit by two major peaks with one in May-October 2020 and the other more deadly in April-June 2021. Surprisingly, the first Indian peak in new cases was reached around the time when the government began lifting nationwide lockdown and focused more on state-level and hot-spot based lock downs [85, 86]; however, there were strict restrictions, such as maintaining social distance and use of face-masks [87]. The second wave struck due to multiple factors and highly-infectious variant-of-concern, also known as SARS-CoV-2 delta variant [23]. A lack of preparation by the authorities in setting up temporary hospitals, shortage of resources such as oxygen and poor management of lockdowns led to major rise of the cases.

There are number of challenges in COVID-19 forecasting due to the nature of the infections, reporting of cases, and effect of lock downs. Nevertheless, despite the challenges and given limited dataset, we have been successful in developing LSTM models for forecasting trend of daily new cases. Our long-term forecasts for two months (October and November 2021) show a steady decline in new cases in India in the respective states. We find that Delhi's two monthly forecasts provide (Fig 11) more uncertainty when compared to Maharashtra and India datasets. We also notice that there is similar level of uncertainty by LSTM and BD-LSTM models for India and Delhi datasets. The major reason the uncertainties are different when you compare Delhi and Maharashtra with rest of India is due to the difference in the trend of cases. In Delhi [15], multiple peaks were observed since the first wave of infections in 2020, whereas in the India dataset, there were only two major peaks. In Maharashtra, a minor peak was observed in November 2021 (Fig 4, Panel a) after the first wave of infections. Hence, the tends captured in the training dataset were relatively different from the states (Maharashtra and Delhi), when compared to India dataset. This suggests that the predictions for

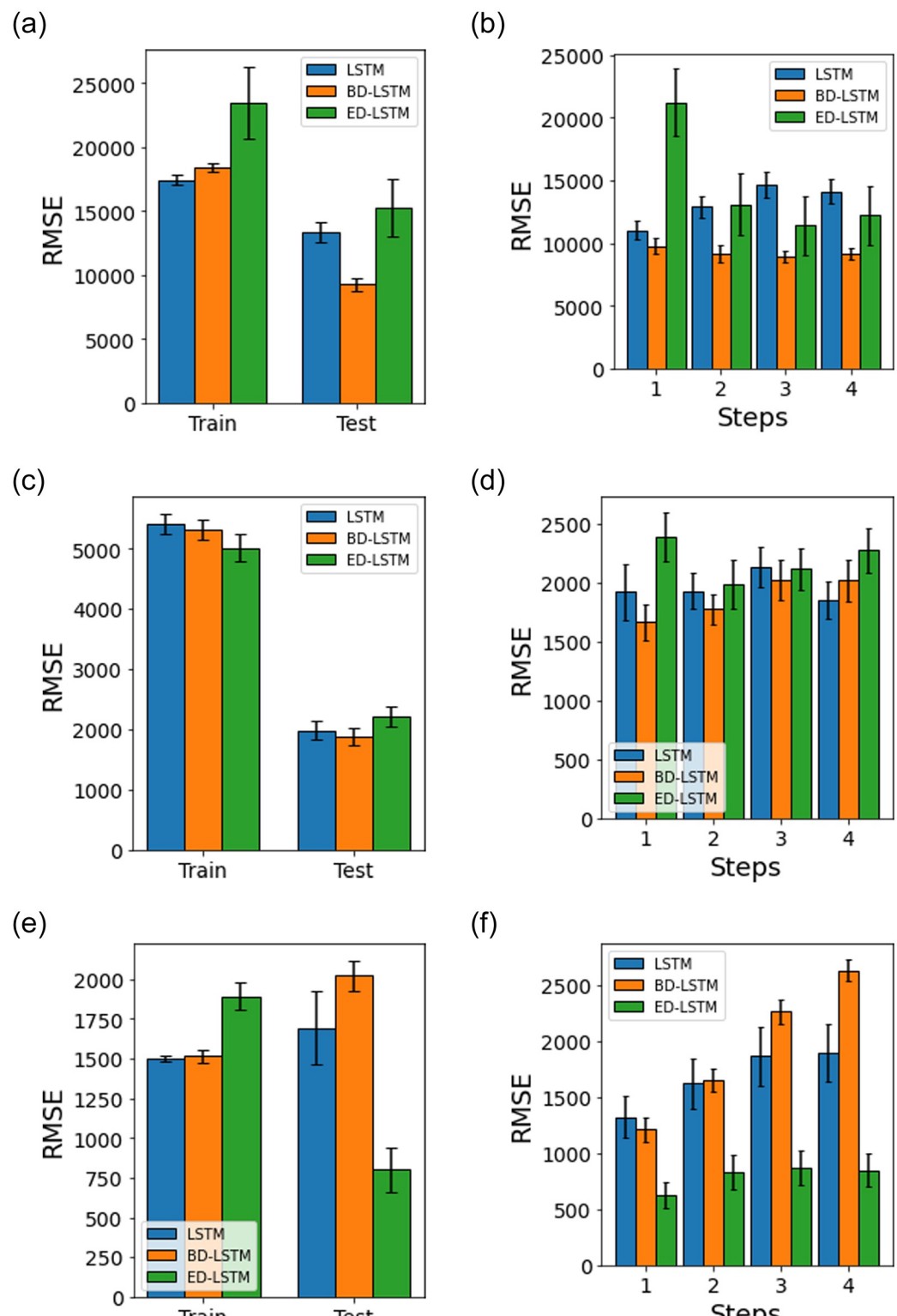

**Fig 9. Multivariate model performance accuracy (RMSE) using LSTM, BD-LSTM and ED-LSTM with a static-split for the train and test datasets, and test prediction horizon (steps).** The error bars represent the mean and 95% confidence interval for 30 experiment runs.

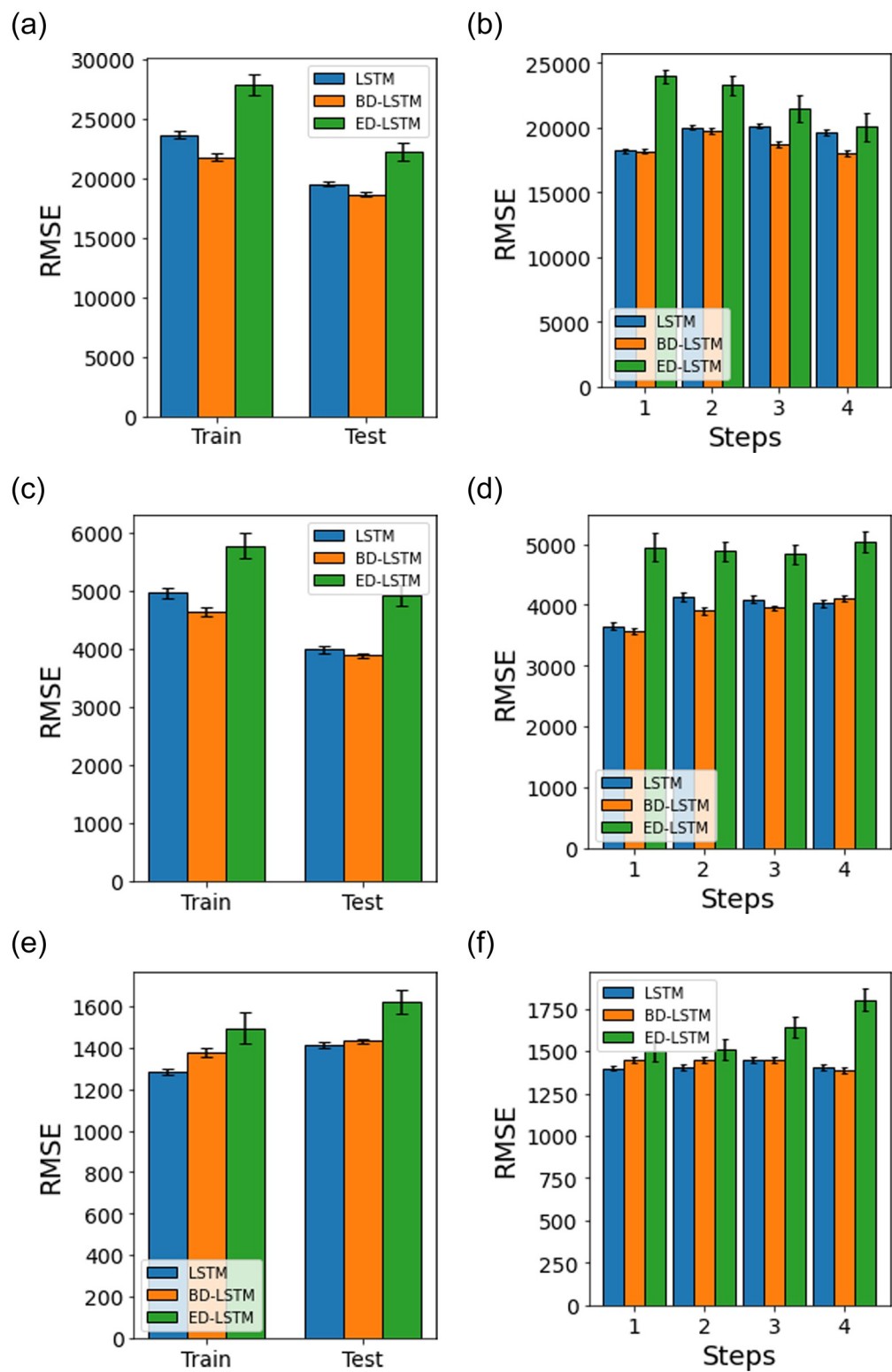

**Fig 10. Multivariate model performance accuracy (RMSE) using LSTM, BD-LSTM and ED-LSTM with a random-split for the train and test datasets, and test prediction horizons (steps).** The error bars represent the mean and 95% confidence interval for 30 experiment runs.

**Table 7. Univariate model performance accuracy on test dataset (RMSE mean and standard deviation for 30 experimental runs across 4 prediction horizons).**

| Model | India | | | | Delhi | | | | Maharashtra | | | |
|---|---|---|---|---|---|---|---|---|---|---|---|---|
| | Random Split | | Static Split | | Random Split | | Static Split | | Random Split | | Static Split | |
| | RMSE | Std. Dev. | RMSE | Std. Dev. | RMSE | Std. Dev. | RMSE | Std. Dev. | RMSE | Std. Dev. | RMSE | Std. Dev. |
| LSTM | 19734 | 1703 | 35540 | 6728 | 1568 | 34 | 266 | 30 | 4762 | 195 | 1925 | 160 |
| BD-LSTM | 21058 | 1746 | 37948 | 6301 | 1628 | 75 | 242 | 22 | 4653 | 214 | 1748 | 195 |
| ED-LSTM | 10732 | 1038 | 62595 | 8932 | 825 | 2 | 435 | 185 | 2365 | 146 | 1594 | 543 |

Maharashtra and Delhi are less certain since they had multiple peaks and outbreaks in the past. The second wave of infections in 2021 began in Maharashtra [88, 89] which has been the state that held one of the most number of COVID-19 cases and novel daily infections (Tables 1–4) [15].

The model uncertainty increases due to the limitations in the dataset and models. Our framework has been limited in capturing social-cultural aspects, population density, and level of lockdowns due to missing information and data. Moreover, inter-state travels and the chaotic nature of spread of COVID-19 infections makes it increasingly harder to provide reliable long-term forecasts. In order to improve forecasting results, the models need to incorporate more features in the data. The model needs to capture features such as travel behaviour, level of lockdowns, compliance in masks and other restrictions, social and cultural lifestyle, local area population density, work and income thresholds, state-wise vaccination rate, and accessibility to information.

We take into account the population density as five Indian cities make the top 50 mostly densely populated cities in the world [90], where Mumbai ranks 5th and Delhi the 40th. The impact of COVID-19 on Indian gross domestic product (GDP) is significant, but not as bad when compared to some of the developed western nations [91, 92]. One of the most crucial aspect of management of spread of COVID-19 infections is the role the government played in timely closing their international borders and enforcing lock-downs to various degrees. We need to note that different countries have different geographical and population dynamics, such as population density and culture. It is not a good idea to compare cities given difference in population density although the overall population may be similar. Overall, it is also important to look at cultural factors such as rituals [93], and role of nuclear and extended families [94]. In countries such as India, there is large portion of inter-state migrant workers [95] and also a large portion of the population is in rural areas [96] that also have extended families. These factors made further challenges in containing the spread of COVID-19 infections and are hard to be captured by computational and mathematical models.

In future work, it is important to incorporate robust uncertainty quantification in collection and sampling data model training and model parameters; hence, Bayesian deep learning framework for COVID-19 forecasts would be needed [97–100]. Moreover, ensemble-based

**Table 8. Multivariate model prediction accuracy on the test dataset (RMSE mean and standard deviation for 30 experimental runs across 4 prediction horizons).**

| Model | India | | | | Delhi | | | | Maharashtra | | | |
|---|---|---|---|---|---|---|---|---|---|---|---|---|
| | Random Split | | Static Split | | Random Split | | Static Split | | Random Split | | Static Split | |
| | RMSE | Std. Dev. | RMSE | Std. Dev. | RMSE | Std. Dev. | RMSE | Std. Dev. | RMSE | Std. Dev. | RMSE | Std. Dev. |
| LSTM | 19524 | 496 | 13325 | 2145 | 1413 | 38 | 1693 | 650 | 3981 | 162 | 1983 | 414 |
| BD-LSTM | 18677 | 521 | 9271 | 1438 | 1449 | 43 | 2021 | 259 | 3888 | 108 | 1886 | 389 |
| ED-LSTM | 22274 | 2174 | 15250 | 6356 | 1621 | 163 | 801 | 394 | 4925 | 503 | 2211 | 478 |

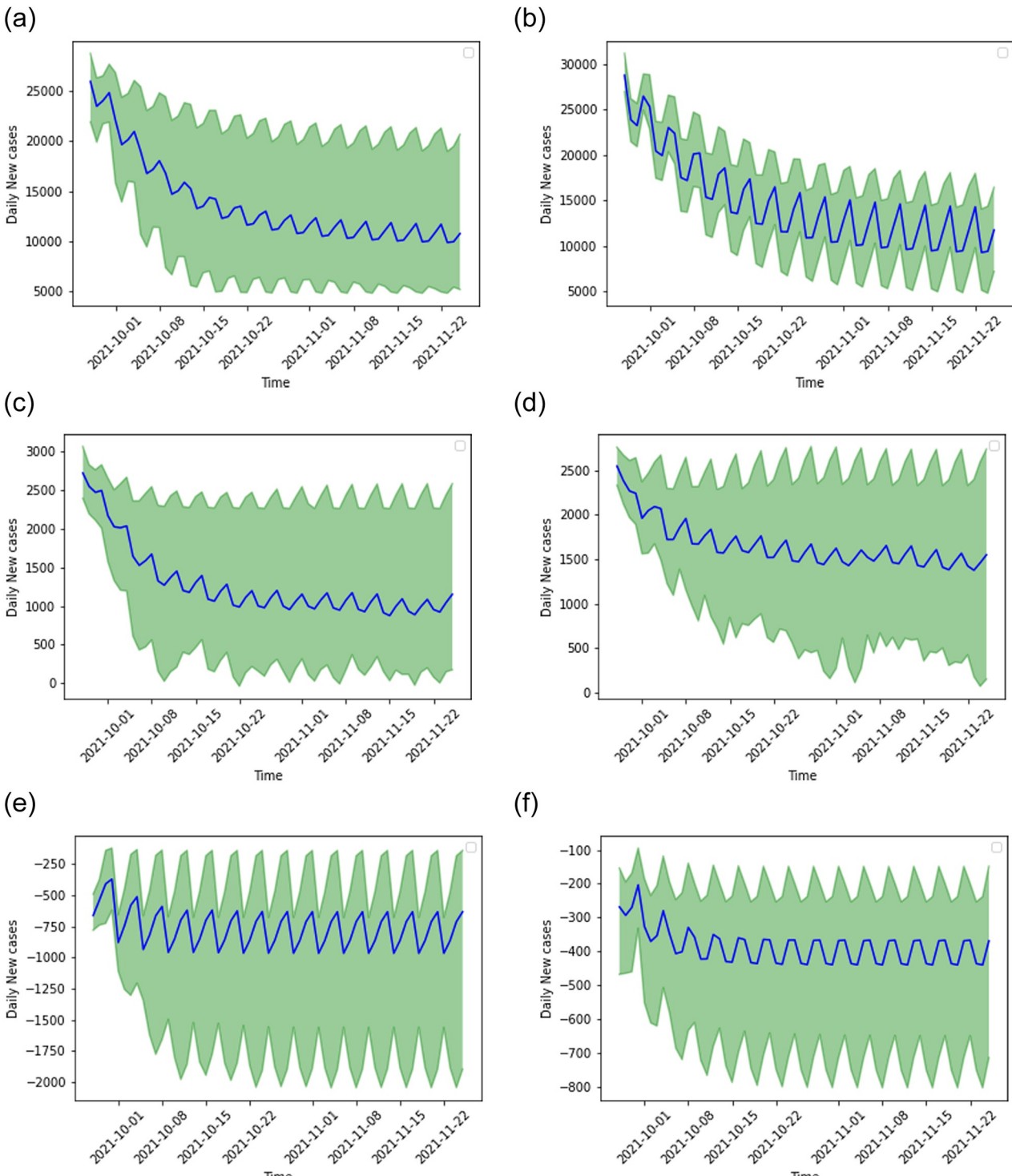

**Fig 11. Recursive univariate LSTM and BD-LSTM models predictions for next 60 days (October and November 2021).** The uncertainty (95% confidence interval shaded in green) and mean prediction is shown in solid black line.

learning methods can be used to combine the different types of LSTM models used in this study. We could also develop similar models for death rate and other trends related to COVID-19. Moreover, deep learning models could be used to jointly model the rise and fall of cases and the effect it has on the economy.

# 6 Conclusions

We presented a framework for employing LSTM-based models for COVID-19 daily novel infection forecasting for India. Our research incorporated some of the latest and most prominent forecasting tools via deep learning, and highlighted the challenges given limited data and the nature of the spread of infections.

Our results show the challenges of forecasting given limited data which is highly biased given that we have two major peaks when considering the pandemic in India. We found that the India and Maharashtra datasets had similar trend in novel cases and model performance. We evaluated univariate and multivariate LSTM-based models with different ways of creating training and test data. The LSTM model variants showed certain strengths and limitations in different scenarios that made it difficult to choose a single model. Generally, we found that the univariate random-split ED-LSTM model provides the best test performance in comparison to rest of the models. Therefore, the data from the adjacent states did not have much effect in the multivariate model since it could not outperform the univariate model. The two months ahead forecast showed a general decline in new cases; however, the authorities need to be vigilant.

## Author Contributions

**Conceptualization:** Rohitash Chandra.

**Investigation:** Rohitash Chandra, Ayush Jain.

**Methodology:** Rohitash Chandra, Ayush Jain, Divyanshu Singh Chauhan.

**Software:** Ayush Jain.

**Supervision:** Rohitash Chandra.

**Validation:** Divyanshu Singh Chauhan.

**Visualization:** Ayush Jain.

**Writing – original draft:** Rohitash Chandra.

**Writing – review & editing:** Rohitash Chandra, Ayush Jain, Divyanshu Singh Chauhan.

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
