## [Decision Letter · Decision Letter 0]

23 Aug 2021

PONE-D-21-19081

Deep learning via LSTM models for COVID-19 infection forecasting in India

PLOS ONE

Dear Dr. Chandra,

Thank you for submitting your manuscript to PLOS ONE. After careful consideration, we feel that it has merit but does not fully meet PLOS ONE’s publication criteria as it currently stands. Therefore, we invite you to submit a revised version of the manuscript that addresses the points raised during the review process.

We look forward to receiving your revised manuscript.

Kind regards,

Chi-Hua Chen, Ph.D.

Academic Editor

PLOS ONE

Journal Requirements:

2. Please include full references for the databases or datasets used, both in the manuscript and the Data availability statement.

Reviewers' comments:

Reviewer's Responses to Questions

**Comments to the Author**

1. Is the manuscript technically sound, and do the data support the conclusions?

Reviewer #1: Partly

Reviewer #2: No

Reviewer #3: Yes

2. Has the statistical analysis been performed appropriately and rigorously? 

Reviewer #1: Yes

Reviewer #2: No

Reviewer #3: Yes

3. Have the authors made all data underlying the findings in their manuscript fully available?

Reviewer #1: Yes

Reviewer #2: Yes

Reviewer #3: Yes

4. Is the manuscript presented in an intelligible fashion and written in standard English?

Reviewer #1: No

Reviewer #2: No

Reviewer #3: Yes

5. Review Comments to the Author

Reviewer #1: In presented form, the authors use three LSTM models to forecast the situation of COVID-19 cases in different cities of India. Only LSTM models are used without any justification why other forecasting models available in theory would fail. This research does not seem to build models out of data, rather an application-based work is presented. The purpose seems to figure out whether LSTM model would succeed in modelling covid-19 data in India. In that case, the authors succeeded. Thus, comparative work is limited in scope. Though lot of work has been presented in literature on covid-19 forecasting (and quoted as well in literature review), but authors fail to simply identify why some prominent ones would have failed on cases from Indian cities.

Minor corrections needed:

1. Grammar needs to be addressed as there are some mistakes in the document.

2. The description after equation (1) does not define ‘g” used in equation (1).

3. The paragraph before Figure 3 states that input sequence is x(t) and output sequence is y(t), but in Figure 3, output sequence y(t) is not stated. Please explain.

4. At the end of section 1, authors state that section 3 presents methodology with data analysis, but section three presents proposed networks only, without data analysis. It is also stated at the end of section 1, that Section 4 presents experiments and results, whereas we only find data collection in the form of Tables and Figures. Please rephrase this paragraph.

Reviewer #2: This manuscript by Chandra et al., aims to develop deep learning models including recurrent neural networks, in particular long short term memory(LSTMs) networks, bidirectional LSTM, and encoder-decoder LSTM models for multi-step (short-term) forecasting the spread of COVID-infections in India. The authors selected Indian states with COVID-19 hotpots and compare with states where infections have been contained and provide two months ahead forecast that shows that cases will slowly decline. According to the authors their approach is promising for long-term forecasts.

My concerns for the manuscript are:

1. The manuscript is not written scientifically. It elaborates the effects of lockdown on economy which is not pertinent to the aim of this manuscript.

2. "India currently (1st December, 2020) has 9,462,809 confirmed cases with 137,621 (1.45 %) deaths which makes the largest in Asia the second highest in the world after the United States. The fatality rate of COVID-19 in India is among the lowest in the world % and steadily declining. India also has one of the fastest recovery rates in the world with 429,753 (4.54 %) active cases, and ranks 8th in the world although 2nd in total cases"

This statement is not true anymore specially with the delta variant and the authors should have revised the manuscript before submission.

3. I couldn't find any description or modeling for the delta variants or emerging covid variants in the manuscript.

Overall I think the manuscript is not suitable for this journal.

Reviewer #3: Authors used different LSTM models to forecast COVID-19 infection in India. The paper is well written and required experiments conducted to validate the results. The following comments may help to improve the paper.

Fig. 2 to Fig. 6: Text in the figures are not visible, increase the font size.

Fig. 11, sub figures (a)-(d) showed high uncertainty compared to (e) and (f), authors needs to comment about this.

Authors may provide more details about LSTM training.

6. PLOS authors have the option to publish the peer review history of their article (what does this mean?). If published, this will include your full peer review and any attached files.

Reviewer #1: No

Reviewer #2: No

Reviewer #3: No

---

## [Author Response · Author response to Decision Letter 0]

20 Oct 2021

Dear Editorial team

The authors sincerely thank the Editor and Reviewers. We have prepared a new version with changes in results and presentation to address the review comments. Please note that the updates are shown in blue in the paper along with our response in files attached.

---

## [Decision Letter · Decision Letter 1]

3 Jan 2022

Deep learning via LSTM models for COVID-19 infection forecasting in India

PONE-D-21-19081R1

Dear Dr. Chandra,

We’re pleased to inform you that your manuscript has been judged scientifically suitable for publication and will be formally accepted for publication once it meets all outstanding technical requirements.

Kind regards,

Chi-Hua Chen, Ph.D.

Academic Editor

PLOS ONE

Additional Editor Comments (optional):

Reviewers' comments:

Reviewer's Responses to Questions

**Comments to the Author**

1. If the authors have adequately addressed your comments raised in a previous round of review and you feel that this manuscript is now acceptable for publication, you may indicate that here to bypass the “Comments to the Author” section, enter your conflict of interest statement in the “Confidential to Editor” section, and submit your "Accept" recommendation.

Reviewer #2: All comments have been addressed

2. Is the manuscript technically sound, and do the data support the conclusions?

Reviewer #2: Yes

3. Has the statistical analysis been performed appropriately and rigorously? 

Reviewer #2: Yes

4. Have the authors made all data underlying the findings in their manuscript fully available?

Reviewer #2: Yes

5. Is the manuscript presented in an intelligible fashion and written in standard English?

Reviewer #2: Yes

6. Review Comments to the Author

Reviewer #2: The authors have made suggested changes in the revised version of the manuscript. I recommend the manuscript for publication in PLOS One.

7. PLOS authors have the option to publish the peer review history of their article (what does this mean?). If published, this will include your full peer review and any attached files.

Reviewer #2: No

---

## [Editor Report · Acceptance letter]

18 Jan 2022

PONE-D-21-19081R1 

Deep learning via LSTM models for COVID-19 infection forecasting in India 

Dear Dr. Chandra:

I'm pleased to inform you that your manuscript has been deemed suitable for publication in PLOS ONE. Congratulations! Your manuscript is now with our production department. 

Kind regards, 

on behalf of

Professor Chi-Hua Chen 

Academic Editor

PLOS ONE